# Algal Bloom Ties: Spreading Network Inference and Extreme Eco-Environmental Feedback

**DOI:** 10.3390/e25040636

**Published:** 2023-04-10

**Authors:** Haojiong Wang, Elroy Galbraith, Matteo Convertino

**Affiliations:** 1Laboratory of Information Communication Networks, Graduate School of Information Science and Technology, Hokkaido University, Sapporo 060-0814, Japan; 2fuTuRE EcoSystems Lab (TREES), Institute of Environment and Ecology, Tsinghua Shenzhen International Graduate School, Tsinghua University, Shenzhen 518055, China; 3Shenzhen Key Laboratory of Ecological Remediation and Carbon Sequestration, Tsinghua Shenzhen International Graduate School, Shenzhen 518055, China

**Keywords:** spatial network inference, biogeochemical networks, predictive causality, bloom prediction, Florida Bay

## Abstract

Coastal marine ecosystems worldwide are increasingly affected by tide alterations and anthropogenic disturbances affecting the water quality and leading to frequent algal blooms. Increased bloom persistence is a serious threat due to the long-lasting impacts on ecological processes and services, such as carbon cycling and sequestration. The exploration of eco-environmental feedback and algal bloom patterns remains challenging and poorly investigated, mostly due to the paucity of data and lack of model-free approaches to infer universal bloom dynamics. Florida Bay, taken as an epitome for biodiversity and blooms, has long experienced algal blooms in its central and western regions, and, in 2006, an unprecedented bloom occurred in the eastern habitats rich in corals and vulnerable habitats. With global aims, we analyze the occurrence of blooms in Florida Bay from three perspectives: (1) the spatial spreading networks of chlorophyll-a (CHLa) that pinpoint the source and unbalanced habitats; (2) the fluctuations of water quality factors pre- and post-bloom outbreaks to assess the environmental impacts of ecological imbalances and target the prevention and control of algal blooms; and (3) the topological co-evolution of biogeochemical and spreading networks to quantify ecosystem stability and the likelihood of ecological shifts toward endemic blooms in the long term. Here, we propose the transfer entropy (TE) difference to infer salient dynamical inter actions between the spatial areas and biogeochemical factors (ecosystem connectome) underpinning bloom emergence and spread as well as environmental effects. A Pareto principle, defining the top 20% of areal interactions, is found to identify bloom spreading and the salient eco-environmental interactions of CHLa associated with endemic and epidemic regimes. We quantify the spatial dynamics of algal blooms and, thus, obtain areas in critical need for ecological monitoring and potential bloom control. The results show that algal blooms are increasingly persistent over space with long-term negative effects on water quality factors, in particular, about how blooms affect temperature locally. A dichotomy is reported between spatial ecological corridors of spreading and biogeochemical networks as well as divergence from the optimal eco-organization: randomization of the former due to nutrient overload and temperature increase leads to scale-free CHLa spreading and extreme outbreaks a posteriori. Subsequently, the occurrence of blooms increases bloom persistence, turbidity and salinity with potentially strong ecological effects on highly biodiverse and vulnerable habitats, such as tidal flats, salt-marshes and mangroves. The probabilistic distribution of CHLa is found to be indicative of endemic and epidemic regimes, where the former sets the system to higher energy dissipation, larger instability and lower predictability. Algal blooms are important ecosystem regulators of nutrient cycles; however, chlorophyll-a outbreaks cause vast ecosystem impacts, such as aquatic species mortality and carbon flux alteration due to their effects on water turbidity, nutrient cycling (nitrogen and phosphorus in particular), salinity and temperature. Beyond compromising the local water quality, other socio-ecological services are also compromised at large scales, including carbon sequestration, which affects climate regulation from local to global environments. Yet, ecological assessment models, such as the one presented, inferring bloom regions and their stability to pinpoint risks, are in need of application in aquatic ecosystems, such as subtropical and tropical bays, to assess optimal preventive controls.

## 1. Introduction

### 1.1. Algal Blooms as the Epitome of Marine Ecosystem Health

Algal blooms are a manifestation of abnormal changes in phytoplankton communities in aquatic ecosystems, such as estuaries and lakes [1,2]. Despite discussions on the perceived global increase in algal blooms attributable to intensified monitoring and emerging bloom impacts, these blooms are increasing worldwide as highlighted from satellite images by Dai et al. [3], and thus they are posing various concerns for the local ecology and global climate. Blooms are highly destructive and persistent [4,5], causing various ecological catastrophes, such as the eduction of vegetated communities, widespread sponge mortality and loss of marine habitat geomorphological structure [6] due to, for instance, habitat calcification [7].

Despite this tremendous damage to aquatic ecosystems, the mutual influence between blooms and the environment has received little attention from scientists and policy makers. Algal blooms are, in fact, the byproduct of nitrogen (N) and phosphorus (P) change but can alter the N/P balance [8] and temperature [9] with implications on carbon sequestration of vegetation in blue carbon ecosystems affected by blooms [10], such as for seagrass. All these elements can be exacerbated by local and global climate change [11].

Despite the limited literature of the effects of blooms into the environment, some studies have explored the relationship between phytoplankton and water quality in bloom conditions [12], climatic and regional variations in phytoplankton as characteristic features of blooms [13,14] and habitat-specific effects that vary by local planktonic biogeochemical stress [15]. Fewer studies have inferred the spatial spreading of blooms characterized as complex networks and predicted blooms based on spatially explicit biogeochemical factors.

This type of biocomplexity study, such as the one we propose, would be necessary to define micro–macro feedback useful for risk assessment, management and policies aimed to minimize eco-environmental imbalances, leading to a decrease in ecosystem health, such as due to blooms. Blooms are the epitome of marine ecosystem health because their emergence is largely related to altered ecohydrological factors at the basin-scale, from land and ocean, leading to quick and persistent increases in phytoplankton with short-term impacts [16] and long-lasting systemic effects on the ecological function and the environment. This is beyond one species or humans only and is related to the progressive degeneration of ecosystem function from its optimality or baseline in relation to the initial or desired conditions.

### 1.2. Complex Marine Ecosystems

Marine microbial food webs consist of heterotrophic protists, phytoplankton, prokaryotes and viruses (i.e., the ocean microbiome). Together, they are responsible for a large part of the production, respiration and nutrient transfer in oceans; they affect, for instance, the carbon cycle both in blue carbon habitats and in the ocean via the carbon pump. As marine ecosystems are increasingly affected by anthropogenic disturbances both from land and ocean, predicting ecosystem responses above critical environmental pressure relies on a better understanding community dynamics, including their composition, spatial/temporal distribution and interactions.

Long-term observations are especially useful for this, and both Galbraith and Convertino [15] and Galbraith et al. [17] provided clear ecological patterns to use as indicators of ecosystem health in relation to ocean microbiome variability intended as a complex network. Chlorophyll-a (CHLa) seems to be the best indicator of community health; however, currently there is the need to quantify how much CHLa variability implies changes in ecological effects (e.g., blooms) and long-term effects, such as on the environment and ecosystem function (e.g., carbon cycle).

Coastal and marine ecosystems that experienced marine heatwaves, which were particularly significant in 2014–2015 worldwide, provide a unique opportunity to study how warming affects community dynamics (namely, microbiome interactions) and how imbalance of the latter affects the environment back in the long term. The presented tool for ecosystemic risk assessment and the results from FL Bay are the main innovations of this paper.

The topological network structure is an effective and intuitive way to describe the dynamical dependencies among diverse of analogous units of an ecosystem, or ecological communities composed of hundreds or thousands of populations of species [18,19]. This is particularly important for marine ecosystems where both structural networks (such as coastal and marine habitat connections and flows) and functional networks (such as prokaryotic and eukaryotic interactions) are not directly visible or known.

Yet, causal network discovery and inference models (e.g., see Li and Convertino [20] for an articulated discussion about ecosystems) are particularly important for mapping the ecological baseline on which current ecosystem assessment and future predictions of ecosystem patterns (tangibly liked to ecosystem services) can be made. Complex networks have great potential to help in solving contemporary real-world problems in a wide range of fields [21,22,23,24,25,26].

Complex networks have been used to analyze the dynamics of pseudo-periodic time series [27] and the functional dynamics of complex systems [28,29,30,31]. Furthermore, networks have become an excellent method for the study of functional and structural dependencies among very complex units with different temporal dynamics [32,33,34,35].

However, most of the considered networks in the literature and particularly those inferred in ecosystems, typically represent relationships based on known or assumed affiliations [36,37] or fixed connections [38]. This makes it difficult to represent the independent local properties of each node and, more importantly, the unique dependencies between different nodes.

This issue is particularly relevant for algal blooms where the biogeochemical networks are hypothesized to vary dramatically over time and space. This has been verified by recent studies on prokaryotic networks whose topological variability was strongly related to systemic ecological stress [15,17]. Nonetheless, no analyses have been made so far on bloom spreading networks, and this research presents a novel template for characterizing and predicting algal blooms based on chlorophyll-a and associated water quality factors.

### 1.3. Ecological Patterns as Chlorophyll-a Spreading Networks

Species, including eukaryotes at the microscale, operate in dynamical ecosystems where the ability to respond to changing environmental flows is paramount. An effective collective response, affecting the re-balancing of optimal ecosystem states requires suitable information transfer among species; thus, ecosystems critically depend on eco-environmental interaction networks. This underpins the process of ecosystem evolution toward low entropy states (characterized by scale-free distribution of CHLa as investigated in this paper) [39] as well as adaptation to new environmental stress states [40], some of which can be undesired, such as those with persistent and large blooms. In this paper, we highlight the central role of information transfer as a salient feature for collective eco-environmental dynamics leading to algal blooms.

Connectomics is broadly defined as the study of structural and functional networks (the connectome), which are maps of a system (such as the nervous system), mainly in the brain; however, this concept has been extended to ecosystems (see Convertino and Valverde Jr. [41]) to characterize both functional species interaction networks, their stimuli with the environment or the envirome itself as set of interdependent environmental processes [15] and habitat networks [20]. The connectome enables understanding of how spreading information is processed (coded, stored, transmitted and decoded in an information sense, which can be any ecological information) at and among different scales of the system (e.g., one node and the whole system, while also considering cross-scale dependencies).

As the connectome is the salient information of ecosystems, its knowledge allows one to improve predictive skills in the short and long term to represent ecosystem patterns. For the aforementioned needs, i.e., to detect the trajectories of spreading blooms and their potential environmental impacts, we demonstrate the capability of an information-theoretic approach to infer bloom networks and biogeochemical feedback. The optimal information flow model was developed initially for inferring species interaction networks in any ecosystem from abundance data [20] and was later applied to predict fish biodiversity patterns Li and Convertino [42] and eco-environmental interactions of the ocean microbiome [17].

The ecological time series underpinning ecological dynamics are particularly important for assessing ecological states and early warning signals of shifts [43] before the inference of ecological networks. The proposed model applies transfer entropy (TE) differences (to target the salient directed interactions) to infer a spatial network strategy that can identify the sources and sinks of bloom outbreak as well as foretell changes probabilistically in water quality factors (in average and fluctuations) when blooms happen.

Through the model, we specifically infer and analyze the spatial ecological corridors determining bloom spread and direct interactions between CHLa and environmental factors to quantify the environmental effects of ecological dysbiosis; previous efforts (see Wang and Convertino [44]) focused on the whole set of biogeochemical interactions useful for forecasting outbreaks, except for bloom spreading networks.

Previously, CHLa has often been used as an indicator of blooms given its sensitivity to environmental changes, ease of monitoring and ability to reflect phytoplankton biomass effectively [45] but has not yet been verified as a systemic indicator of ecosystem health related to ecosystem function. We discuss the results of applying this model to algal blooms observed in Florida Bay (Florida, USA) in the Florida Everglades National Park between 2005 and 2006 when a recurrence of large blooms was observed.

Due to its unique lagoon configuration and climate, Florida Bay (Figure 1) regularly experiences algal blooms [46] as frequently as many other aquatic ecosystems in subtropical and tropical climates. Thus, for algal blooms, there is the need for a powerful dynamic prediction model to support decision-making and bloom prevention.

## 2. Materials and Methods

The proposed TE network inference model that can be used for variable interaction discovery and prediction at multiple scales is explained. Its structure is graphically shown in Figure 2. The model is a further refinement on the one proposed by Li and Convertino [20] for the use of TE differences to prune the network and define salient predictors of ecological patterns, which are algal blooms in this case.

### 2.1. Datasets

The Florida International University Southeast Environmental Research Center (FIU SERC) established a water quality monitoring system of 28 spatially distributed stations in Florida Bay (Figure 1), where each station (considered as a node in a network perspective) collects monthly data on chlorophyll-a (CHLa), total organic carbon, inorganic and organic nitrogen and phosphorus (TN and TP), turbidity (TURB), pH, salinity (SAL), water temperature (TEMP) and dissolved oxygen (see Boyer and Briceno [47] and Nelson et al. [14] for a description on how data are measured).

We used a threshold-based quantile regression method (analogous to Nelson et al. [14]) to establish an average threshold of ≥2 μgL^−1^ on CHLa, universally applied to all stations, to distinguish bloom from non-bloom states across all stations. Initially, the dataset for this study spanned 2004 to 2006, corresponding to before, during and after a severe bloom outbreak in Florida Bay in 2005 [48] in terms of a CHLa extreme. In 1999, several blooms were observed in the same area but with lower CHLa extremes [14].

Then, the dataset (comprising all 2004, 2005 and 2006 CHLa monthly data) was filtered to include only those months and stations with CHLa values exceeding the critical blooming threshold, i.e., those months and stations indicating sustained bloom conditions. As a result, the final dataset contained 18, 63 and 136 rows of measurements (i.e., months) for the 2004, 2005 and 2006 bloom periods (pre-, peri- and post-bloom), respectively. More generically, 2005 can be considered as epitomic of bloom outbreaks, while 2004 and 2006 are representative of early and post-bloom periods.

### 2.2. Ecosystem Organization and Connectome

The entropy of the ecosystem, manifesting ecological disorganization in relation to CHLa variability, is dependent on the probability distribution functions (pdfs) that affect TE calculated on the pdf divergence and asynchronicity. The TE variability of an area, or the whole system, can be decomposed into eco-environmental interactions (considering CHLa and environmental factors acting as determinants or effect of ecological imbalance) and the ecological areal interactions underpinning bloom spread. This variability affects the organization propagation of CHLa (i.e., how randomly distributed CHLa is) and, in an information-balance equation, can be written as the spatio-temporal convolution of the aforementioned components composing the ecosystem connectome, i.e.,
(1)$$\begin{array}{c} \overset{\mathrm{eco}-\mathrm{function}}{\overbrace{H(\mathrm{CHLa})}}=\sum_{m,n}\int_{0}^{t}\overset{\mathrm{eco}-\mathrm{connectome}}{\overbrace{\underset{\mathrm{eco}-\mathrm{env}\mathrm{feedback}}{\underbrace{\left(1-TE\left(X_{m},{\mathrm{CHLa}}_{m}\right)\right)}}*\underset{\mathrm{eco}-\mathrm{corridors}}{\underbrace{\left(1-TE\left({\mathrm{CHLa}}_{m,n}\right)\right)}}}}\;d\tau \;, \end{array}$$
where *X* stands for all other environmental variables except for CHLa, and m,n stands for the location of each area being monitored over the period *t*. The specific TE chosen in Equation (Equation 1) is related to TE analytics and the posed objectives, to be later specified. It should be noted that the time delay τ between eco-env factors in Equation (Equation 1) has been set to one due to the sub-monthly variability of CHLa and the resolution of the data.

Equation (Equation 1) is focused on CHLa patterns where networks are the backbone determinants of the ecological “weave” (CHLa interconnected patterns) that can be potentially controlled. Space and time are the dimensions along which CHLa is considered, plus other dimensions along gradients of environmental features on which stress–response patterns and related features (e.g., early warning signals and risk thresholds) can be derived. The networks define sources, sinks, pathways and determinants to guide monitoring and environmental control for bloom prevention.

In this paper, we specifically analyze the spatial ecological corridors determining bloom spread and direct interactions between CHLa and environmental factors (second and first term in Equation (Equation 1), where, for the latter, only ${\mathrm{CHLa}}_{m}\to X_{m}$ interactions are considered) to quantify environmental effects of ecological dysbiosis; Wang and Convertino [44] focused instead on the whole set of biogeochemical interactions useful for forecasting except for bloom spreading networks.

### 2.3. Eco-Environmental Network Inference

Transfer entropy (TE), constructed from information entropy [49], measures the causal relationship between two asynchronous and divergent variables (expressed as a time series) *X* and *Y* (in the bivariate form, yet not accounting for second-order indirect interactions) by quantifying the predictive information flow between them [50]. Previously, the TE-based model, called the optimal information flow model (in relation to the maximization of ecosystemic entropy to gather the largest information), was used to discover causal relationships in human and aquatic ecosystems, e.g., for bacteria [15,17,25] and fish interactions [20] and to assess ecosystem health.

The information flow, and thus the predictive relationship between variables, is bi-directional. In this paper, we took the form of bivariate TE (while skipping interactions higher than the third-order, which was our first modeling assumption considering the weakly third-order interactions between environmental factors [51]) and calculated the difference between the pairwise information flows to identify the strongest causal factor, i.e., $TE_{X\to Y}$ and $TE_{X\leftarrow Y}$ (where *X* and *Y* can be either ecological, such as CHLa, or environmental variables) as follows:(2)$$\begin{array}{cc} {TE}_{X,Y}\; & ={TE}_{X\to Y}-{TE}_{X\leftarrow Y}\\ & =\sum p\left(y_{t+1},y_{m},x_{n}\right)log\frac{p(y_{t+1}\left|y_{m},x_{n}\right.)}{p(y_{t+1}\left|y_{m}\right.)}\\ & -\sum p\left(x_{t+1},x_{n},y_{m}\right)log\frac{p(x_{t+1}\left|x_{n},y_{m}\right.)}{p(x_{t+1}\left|x_{n}\right.)}, \end{array}$$
where $x_{t+1}$ and $y_{t+1}$ are the values of variables *X* and *Y* at time t+1 (yet, Δt=1 month, which is our second modeling assumption considering the fact that CHLa values are very sensitive to past changes in the immediate past reflecting Markovian dynamics [51], while long-term increasing trends can lead to extreme CHLa shifts). $x_{n}$ and $y_{m}$ denote the histories of time-varying variables *X* and *Y* up to t−n+1 and t−m+1, respectively. $TE_{X\to Y}$ is the transfer entropy of time series variable *X* to *Y*, whereas $TE_{X\leftarrow Y}$ indicates the transfer entropy of *Y* to *X*.

In this study, we considered only positive $TE_{X,Y}$ where X= CHLa and *Y* are all other environmental factors for eco-environmental feedback in Equation (Equation 1) and considered all $TE_{X,Y}$ where *X* and *Y* are both CHLa in two different nodes. Additionally, in the TE calculation, we did not investigate the optimal time delay between *X* and *Y* nor the optimal set of factors that are predictive of CHL, as in [20], due to: (i) the fact that bloom eco-env feedback occurs at scale lower than one month (at which data are available) and (ii) our interest into the entire systemic dynamics. This first part of all TE inference is considering all pairs of variables (Figure 2A).

The unbounded causality matrix, or more precisely the predictive causality matrix **TE** unconstrained to any prediction of biodiversity patterns as in [42], based on calculated TEs without the optimization of Δt and predictive environmental factors of ecological patterns in an optimal information flow perspective, can be constructed as follows:(3)$$\mathrm{TE}=\left[\begin{array}{ccc} TE_{1,1} & \cdots & TE_{1,Y}\\ \vdots & \ddots & \vdots \\ TE_{X,1} & \cdots & TE_{X,Y} \end{array}\right].$$
where $TE_{X,Y}$ is indeed a difference of transfer entropies as in the transfer entropy graph neural network model (TEGNN) (originally developed by Duan et al. [52] and applied to algal blooms by [51]) in contrast to the optimal information flow model (OIF) originally developed by Li and Convertino [20]. For each year, two networks were constructed with each defined by an underlying matrix of transfer entropy differences TE. One inferred matrix was a spatial network in which the 28 stations were nodes, and the causal influences among them were the edges.

The time series used to calculate the transfer entropy differences (Equation (Equation 2)) in this network were the time series of CHLa measurements at each station. The second inferred network was a water quality network, in which the nodes were the water quality factors (CHLa, TN, TP, SAL, TEMP and TURB), and the edges were the causal influence among them. In this study, however, the causality matrix underlying the water quality network was further filtered to focus only on the effect of CHLa on other water quality factors in relation to the objective to quantify this eco-environmental feedback; the reverse effect of water quality factors on CHLa and the interactions among water quality factors were not considered in this task but were in Convertino and Wang [51] and Wang and Convertino [44].

Following the Pareto principle [53], the largest 20% of values (and not 20% of events) identify the most influential variables (stations or factors), which are Pareto elements with the largest portfolio effect (à la Anderson et al. [54], which is about variance-mean scaling related to CHLa distributions), yet defining the risk of blooms. Therefore, we only retained the largest 20% of values in the mTE in order to focus on the most influential variables.
(4)$$\begin{array}{c} mTE_{X,Y}=\left\{\begin{array}{cc} TE_{X,Y}, & TE_{X,Y}>d\\ 0, & \mathrm{otherwise} \end{array},\right. \end{array}$$
where *d* is the threshold value, or Pareto critical value, of significant causal relationships as necessary and sufficient to predict bloom dynamics. If $TE_{X,Y}>d$, then *X* is a significant cause of *Y*; otherwise *X* is a weak cause of *Y*, or *X* and *Y* are mutually causal without any preferential direction in causality (Equation (Equation 4) and Figure 2). Additionally, the values in the matrices for each of the three years were normalized to enable better comparison of the inferred interactions. This second part of salient TE selection is about the network pruning (Figure 2B).

### 2.4. Eco-Environmental Factor Predictive Causality

The total outgoing transfer entropy (*OTE* as in Galbraith et al. [17] reflecting the total direct influence of one variable for all other influenced variables) of a node can be used to measure its influence on the collective dynamics (Equation (Equation 5)) as follows:(5)$$\begin{array}{c} OTE=\sum_{Y}mTE_{X,Y}, \end{array}$$
where *OTE* is the sum of the *X*th row in the mTE matrix (after the application of the threshold as in Equation (Equation 4). The larger the *OTE*, the stronger the influence of node *X* on all other directly connected nodes in the network. In an information-balance perspective, *OTE* is the cumulative effect of all environmental or ecological variables for a node.

## 3. Results and Discussion

### 3.1. Spatio-Temporal Spreading and Fluctuations

To infer and characterize the spreading networks of blooms, while underpinning the ecological risk, we considered Florida Bay blooms between 2004 and 2006. We inferred a novel spatial influence network underpinning bloom spread among a set of spatially distributed water monitoring stations. This was achieved by deriving a TE matrix from spatio-temporal patterns of CHLa derived from monitored stations (see Section 2.3). The TE matrix for 2004 suggests that the study site was free of severe blooms, except for a few stations in the northwest: specifically, stations 16, 14, 25 and 26 (Figure 1) at least in 2004 where the resurgence of blooms was observed after the large bloom in 1999 [14,51].

Ecological spreading corridors are defined by the most divergent and asynchronous CHLa among nodes, while defining the most likely interdependent area, at least in a predictive causality sense (causality considering all other feasible connections, which are all other nodes in this case). Divergence and asynchronicity, as highlighted by Li and Convertino [20], are related to the difference in pdfs of CHLa (in two nodes) at different or equivalent time periods, respectively.

Spreading can be related to marine currents; however, in this study, the purpose is not to define the precise mechanisms underpinning the ecological patterns but rather to define the patterns’ backbone networks, which are the salient spreading networks. This also identifies the potential coastal areas of influence of biogeochemical loads in FL Bay and the maximum extent of blooms—something that is poorly quantified but necessary for bloom prevention.

In analogy, runoff in terrestrial basins are predicted equivalently to CHLa, where the amount (and distribution) of water in different locations changes in an asynchronous way and is dependent on river network spreading to define the timing and divergent volume. True causality, leaving aside the feasibility of its assessment, must be included considering all areas where CHLa can spread, which, in a bay, is virtually everywhere; however, this is challenged by the data limitation that is constrained only to the used stations in this case.

The properties of network edges, representing eco-environmental interactions, depend on mTE (Equation (Equation 4)). The edge directions of the spreading network (Figure 3) suggest that an algal bloom would have initiated around station 16 and then moved west with a preferential direction toward the northwest. The edge colors (proportional to mTE) suggest that the bloom was moderately strong but localized in 2004 (Figure 3A) with a high probability to continue growing in the bay (due to mTE directions). The spatial spreading mTE matrix for 2005 revealed large and widespread bloom outbreaks that were concentrated in the western and central areas of the bay (Figure 3B).

In that year, the spatial influence was the strongest near stations 25, 16, 14 and 12 in the northwest and station 28 in the south region. The edge colors indicate that the bloom at all stations was moderately strong and also very likely to continue in the NE direction. Station 28 seems largely affected by many other stations in the bloom spread and yet is likely a sink node with potentially strong ecological effects also considering its proximity to the FL coral reef.

The matrix for 2006 (Figure 3C) shows the most extreme area interactions as well as a reversal in the spreading of blooms, i.e., moving from NE to central areas. The edge colors imply that bloom activity was extremely high, covering a wide area of Florida Bay. Nonetheless, the resulting graph suggests that, after the largest outbreak, the bloom moved from the easternmost into the north-central area, while the bloom in the west region dissipated.

We show how, by analyzing the information flow among spatially distributed nodes, it is possible to model the spatial spread of a phenomenon, such as algal blooms. In addition, this approach is able to detect sources, sinks, directions and salient pathways of bloom spreading. Due to various unaccounted factors, such as wind intensity and direction, current direction and bathymetry, there is a certain dynamic spatial change of blooms that is not attributed to the aforementioned factors. However, the model can take into account any environmental factor if available and can attribute the degree of variability of CHLa.

In a complex network sense, the bloom spatial network in 2004 is small in scale and regular in topology but has an obvious active station (station 16) that is an actively connected hub for bloom spreading. Therefore, it is much easier to take measures against blooms at this time (whether possible) or to prevent triggers by controlling environmental determinants. This area is well-know to be heavily influenced by nutrient efflux from the Everglades [14].

Particularly with the outbreaks of blooms in 2005 and later in 2006, the network has many more areas that are very active and affected, yet bloom management becomes more difficult. Over time, a spreading network transition is observed from a regular/small-world in 2004, to scale-free in 2005 and regular (or uniform) topology in 2006 with long-range connections.

### 3.2. Water Quality Trends and Bloom Impacts

The investigation of the impact of CHLa extremes (the magnitude, duration and frequency) on ecosystem health is a poorly covered topic in science. To explore how algal blooms impacted the water quality in Florida Bay, we analyzed how CHLa impacted other water quality variables using TE (see Section 2.3). We focused our analysis on how CHLa implicated potential changes in water quality—in terms of predictive causality—for stations where extreme blooms were most likely.

At the most active station in 2004 (i.e., station 16, characterized by coastal marshes, which is likely the source of blooms; see Figure 3A), blooms did not affect TN, TP, SAL and TURB, except for a slight effect on the water temperature (see Figure 4A). Rather, TN, TP, SAL and TURB, likely driven by a riverine efflux in the bay, triggered CHLa changes leading to blooms as highlighted in Wang and Convertino [44] and Convertino and Wang [51]. In 2005, the impact of blooms on other water quality factors was mostly evident at station 25, which is a deep-water mangrove habitat, where the blooms were the most intense (see Figure 3B).

CHLa induced not only water temperature changes but also variations in the total nitrogen and salinity (TN and SAL) with a higher impact on the latter (Figure 4B). In 2006 (see Figure 3C), where blooms were the most extreme but localized (the NW area), the effect of blooms on water quality peaked at station 3, followed by stations 5 and 2 and then station 6 in terms of magnitude (Figure 4C). Stations 2 and 3 experienced blooms throughout the year, while station 6 had a relatively short bloom (7 months as reported in the data).

At stations 3 and 6 (characterized more by tidal flats), blooms induced changes in the water temperature, salinity, total phosphorus and turbidity, while, at stations 2 and 5 (characterized more by submerged marshes), blooms led to substantial fluctuations in the total nitrogen, total phosphorus, salinity and turbidity. Information flow patterns (TE patterns) suggest that blooms first strongly caused water temperature alterations, then enhanced the salinity and nitrogen and later impacted other nutrients (phosphorous) and the turbidity. This is aligned with an understanding of the underlying microbiological processes [51].

In the vicinity of station 2, blooms caused a large change in salinity, while the effects on TN, TP and TURB were less significant. As blooms are a manifestation of eutrophication in water bodies, large amounts of phytoplankton cause dramatic changes in the total phosphorus and turbidity, such as near station 3, with a minor influence on the temperature and salinity. Around station 5, the bloom had a strong influence on the turbidity and salinity, with a minor impact on the TN and TP due to the deeper water in this area.

Despite the bloom near station 6 being relatively short, it still caused elevated changes in both the salinity and turbidity and, in a minor way, in the water temperature and total phosphorus. In general, the occurrence of blooms had serious effects on the total phosphorus, salinity and turbidity in the eastern zone of Florida Bay; a worrisome condition because of the highly valuable biodiversity in that area comprising a wide set of sponge, fish and coral species.

Our results reveal that algal bloom severity also caused environmental degradation a posteriori beyond the direct causal effects of environmental change (particularly from temperature in the ocean and nutrients from estuarine efflux) in triggering blooms a priori. Certainly, the primary causal pathway is about temperature leading to CHLa changes; however, the inferred networks also manifests the feedback of CHLa change on temperature. While this can be minor with respect to the first mechanism, it is also possible in relation to algal overgrowth and local temperature increase.

This substantiates environmental changes due to ecological imbalances [9], such as the oceanic positive feedback mechanism, which can lead to further increases in phytoplankton growth, chlorophyll-a concentration and temperature. Blooms are ecological processes that consume energy and yet increase the local temperature—precisely, algal blooms absorb light from the sun and carbon from the atmosphere, which increases the temperature of surface water. Whether this can be captured by our data or other data is an open question, but what is certainly true is that the bidirectional CHLa-temperature feedback is inferred as well as the CHLa-salinity.

Rising temperature, also related to local eutrophication, implies more evaporation from waterbodies and yet higher salinity if the hydrology is not changed. Of course, if the algae grow overly much (in term of biomass), a large amount of oxygen is depleted when they die, and this creates hypoxia and cascading risks, such as the death of species and the emergence of toxins. This can also lead to an exceeded capacity of zooplankton to sink carbon to the bottom of the ocean and, thus, an increase in the size and frequency in blooms, which is not good for the generated temperature, which is a co-occurring risk factor.

### 3.3. Bloom Intensity and Area Dependency

We explored the interaction dynamics of blooms by analyzing the annual probability distribution, or pdf, of the outgoing transfer entropy (OTE; see Figure 5) pre-, peri- and post-bloom. *OTE* quantified the extent to which blooms around one area can predict CHLa dynamics (in terms of the value and distribution) in other areas: higher *OTE* values indicate higher area interactions with higher spreading and predictability. In 2004, the *OTE* ranged between 0 and 1.7; most values with a non-zero probability were between 0 and 0.6.

It can be seen that most of the stations have no bloom, resulting in a low probability of large values of *OTE* but a high probability of low *OTE*. The pdf is bimodal with a leptokurtic character. In an ecological sense, the dynamics are characterized by highly localized blooms and few traces of bloom emergence in other areas. Thus, the bloom spatial network system was relatively contained in 2004 and corresponds to a regular/small-world topology (Figure 3). This also corresponds to simple low-TE dynamics of eco-environmental interactions (Figure 4).

In 2005, the *OTE* range increased to a maximum of 4.0, with most OTEs having a higher probability than in 2004. In 2006, the range of OTEs increased even further to a maximum of 13, with all *OTE* values having a higher probability. This also corresponds to a shift in the pdf to being more platykurtic while highlighting more widespread and common bloom dynamics.

From the perspective of complex networks, the number of nodes with large *OTE* values increased over time. This indicates an explosive spread of blooms across FL Bay. Therefore, the initial energy dissipation became higher over time. In 2005, the system was in an active and complex state, which makes the management of blooms extremely challenging. The 2006 pdf has higher entropy because it is a distribution closer to a Poisson pdf than previous years.

The pdf of the OTE proves that OTE reflects the probabilistic state of ecosystems with particular reference to algal blooms in this case. The higher the entropy, the larger the effect of blooms and the higher the ecological effects; interestingly, for FL Bay, we notice that, the higher the entropy, the more scale-free the bloom spreading network is, although a time delay may exist between ecological effects (CHLa, which is more random, such as in 2006) and the largest spreading network (which was in 2005) signifying potential long-term effects.

By flipping the pdf, it is possible to obtain information on the ecosystem potential landscape about the energy dissipation, likelihood of shifts and relative stability of bloom conditions (Figure 5). The energy dissipation of the system, which is the potential amount of energy consumed by ecological processes, is visualized, where ∝max[p(OTE)]−p(OTE), which scales with ∼1/p(OTE); therefore, the higher the leptokurtic character of the pdf is, the lower the energy dissipation (such as in 2004).

The energy potential also gives the number of ecological states (metastable states are identified by the point where the pre- and post-curvature of the energy landscape diverges in sign; these are represented by the balls in Figure 5), the probability of a configuration to be stable (the lower the energy potential with respect to all other states is, the higher the stability) and the likely shifts among states (proportional to the slope of the energy potential), all of which define the ”resilience” of the ecosystem, which is the rapidity to bounce back to initial states.

Higher entropy corresponds to higher energy dissipation in relation to larger and more random OTEs. This implies a lower probability of CHLa stable states, which are much closer to each other and increase in number, implying a higher likelihood of shifts with larger ecological impacts. For FL Bay, the energy dissipation also increases in average value for the pre-, peri- and post-bloom periods indicating a diminishing resilience and loss of complexity of the system; this also highlights the persistent effects of blooms despite their relatively short duration.

## 4. Conclusions

This study uniquely proposed a model based on optimal transfer entropy (TE) with TE differences to infer bloom spatial dependencies, which were used to pinpoint risk areas and pathways to target monitoring and controls. Blooms showed non-trivial spreading patterns manifested by network transitions with different stability results that determined their persistence and potential ecological effects. For FL Bay, we predominantly highlighted the spatial trends and the neglected impacts of algal blooms on the water quality. The following specific results are worth highlighting.

We showed how CHLa patterns carry information regarding the underpinning ecohydrological networks (and associated spreading determinants, such as nutrients) that support ecosystem function and services. Salient Pareto interactions were defined via thresholding TE differences with a threshold of causal significance that was set to consider the top 20% of TEs (related to the tail of scale-free CHLa probability distribution function), i.e., necessary and sufficient interactions to predict the risk of bloom spreading.More generally, the discovery and inference of the ”ecosystem connectome” (as biogeochemical determinant and spreading networks) allows for the assessment of ecosystem health (quantified by the proximity to an optimal condition, such as the non-bloom state) as well as the investigation of causal determinants and their sources, proximity to ecosystem shifts and targeted ecohydrological controls.Through spatial analysis of bloom spreading networks, we showed how regions not previously involved in blooms (i.e., the highly biodiverse NE tidal-flat habitats with corals and sponges) were caused by large imbalances of CHLa in the western and central blooms, which were causally involved. The latter regions were characterized by CHLa that was more randomly distributed and a higher probability of CHLa extremes. This probabilistic structure, reflecting the spatial distribution of CHLa, is likely tipping eastern regions to similar bloom endemics. From the perspective of complex networks, this bloom event (2004–2006) evolved from a spatial network with a localized trigger area and a small-world topology to a random topology with long-range spatial diffusion.In 2005, when most stations were blooming, the spatial spreading network was scale-free (theoretically optimal in a purely topological and predictive sense [55,56]) with a random biogeochemical network, including CHLa (topologically suboptimal), which underpins the dichotomy between structural and functional networks for ecological risks.In terms of temporal dynamics, subsequent to the first bloom outbreak, persistent and recurring blooms were observed for several NE areas with long-lasting environmental impacts on turbidity and salinity aggravated by temperature increases. Bloom sources were related to central coastal marshes and, to a lower extent, mangrove habitats. We further showed that blooms were a recurring and persistent phenomenon over a long period of time with continuous outbreaks in interdependent regions. This led to higher energy dissipation and larger instability dictated by the more random distribution of CHLa, which was associated with a more uniform network with long-range connectivity regardless of habitats, likely leading to the loss of ecological heterogeneity.The analysis of biogeochemical factors affecting water quality showed that the occurrence of blooms could only affect small fluctuations of temperature at the beginning of the blooms; however, repeated bloom outbreaks largely affected other biogeochemical factors (such as salinity, turbidity and CHLa triggering hysteresis or memory effects) that are poorly systemically controllable due to the loss of vegetation and other keynote species.The concentration of CHLa can be influenced by temperature and salinity, and changes in the CHLa concentration can, in turn, have indirect effects on water temperature through various ecological processes. In some regions, facilitated by shallow-water habitats, a water temperature increase can stimulate phytoplankton growth and increase the concentration of CHLa. The increased CHLa can, in turn, absorb more sunlight, which can lead to local warming of the water.In the long term, the persistence of blooms, i.e., high CHLa, may also alter nutrient cycling as highlighted by other studies with the term “oceanic positive feedback mechanism” [11], and our model was able to infer this secondary causal pathway together with the primary one, where temperature change led to CHLa change and blooms. This underscores that bloom management should start from the source, otherwise blooms’ environmental impacts will gradually expand and become uncontrollable, thus, also affecting the ecosystem stability and resilience and settling into undesired ecological states.

Although the intensity, duration and spatial distribution of blooms are governed by a multiplicity of factors, CHLa variability (independently of any trigger) still has a wide degree of predictability and control in an ecosystem perspective considering both predictive and ecological engineering models. We proposed a data-based inferential model to be used for ecological intelligence to look into patterns of risk (source and pathways), trajectories and determinants.

Our proposed spatial and biogeochemical network inference model provides valuable information for the forecasting and management of blooms—for instance, by pinpointing monitoring and nature-based solutions in source areas, such as coastal blue-carbon habitats to inhibit progressive eco-environmental imbalances and the related impacts. Further work will look into the precise quantification of critical thresholds (habitat- and climate-specific or universal) as early warning signals of environmental factors (including controls) that lead to persistent blooms and accounting for systemic stress, that is reflected by the condition of habitats as their ecological history.

## Figures and Tables

**Figure 1 entropy-25-00636-f001:**
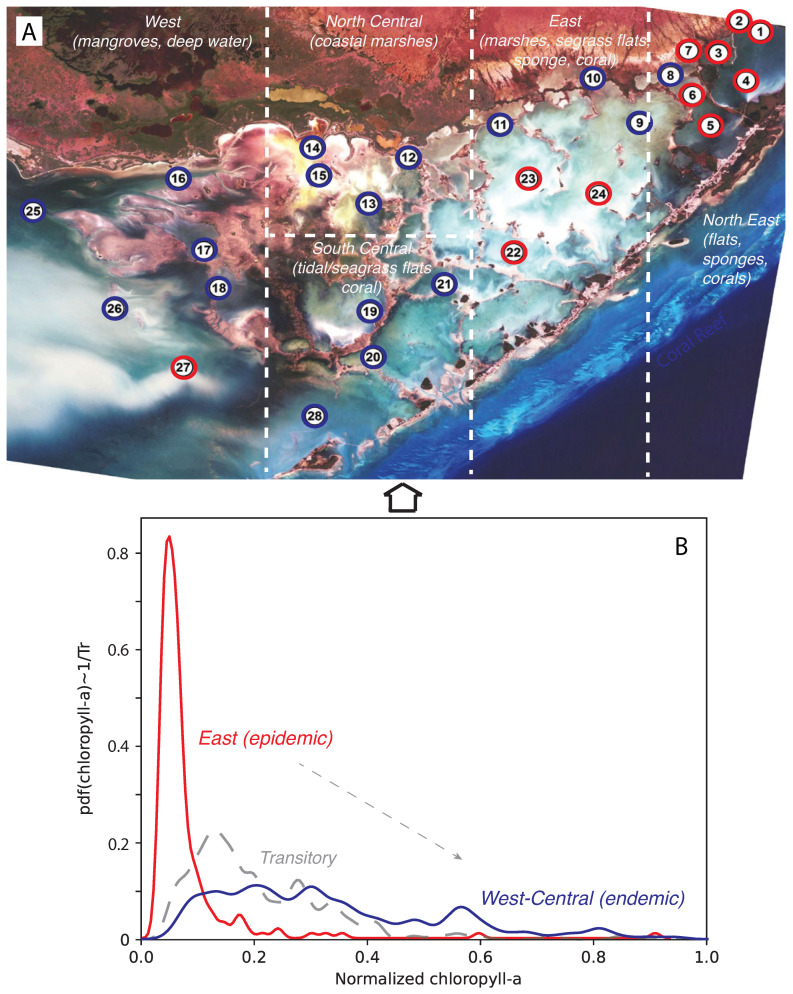
Florida Bay and area classification based on CHLa dynamics. The red–blue classification in plot (**A**) is related to the probabilistic structure of CHLa as highlighted in plot (**B**). Plot A also highlights the main habitats and species present in FL Bay.

**Figure 2 entropy-25-00636-f002:**
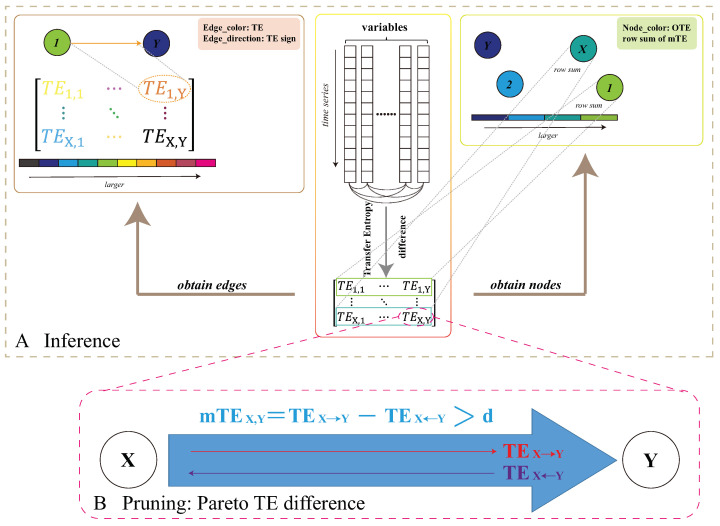
Ecological corridor inference model. The structure of the TE inference model. Here, variables are annotated as *X* and *Y* generically. *X* can be thought of as CHLa and *Y* as all other environmental variables. The first step of the proposed model is to infer variable pairwise interaction as TE and node collective influence (*OTE*), determined via Equations (Equation 2) and (Equation 5), respectively. The second step is to prune the network considering only salient Pareto interactions via thresholding TE differences with a threshold *d* of causal significance, which is set to consider the top 20% of TEs (Equation (Equation 4)) that are necessary and sufficient to predict bloom spread.

**Figure 3 entropy-25-00636-f003:**
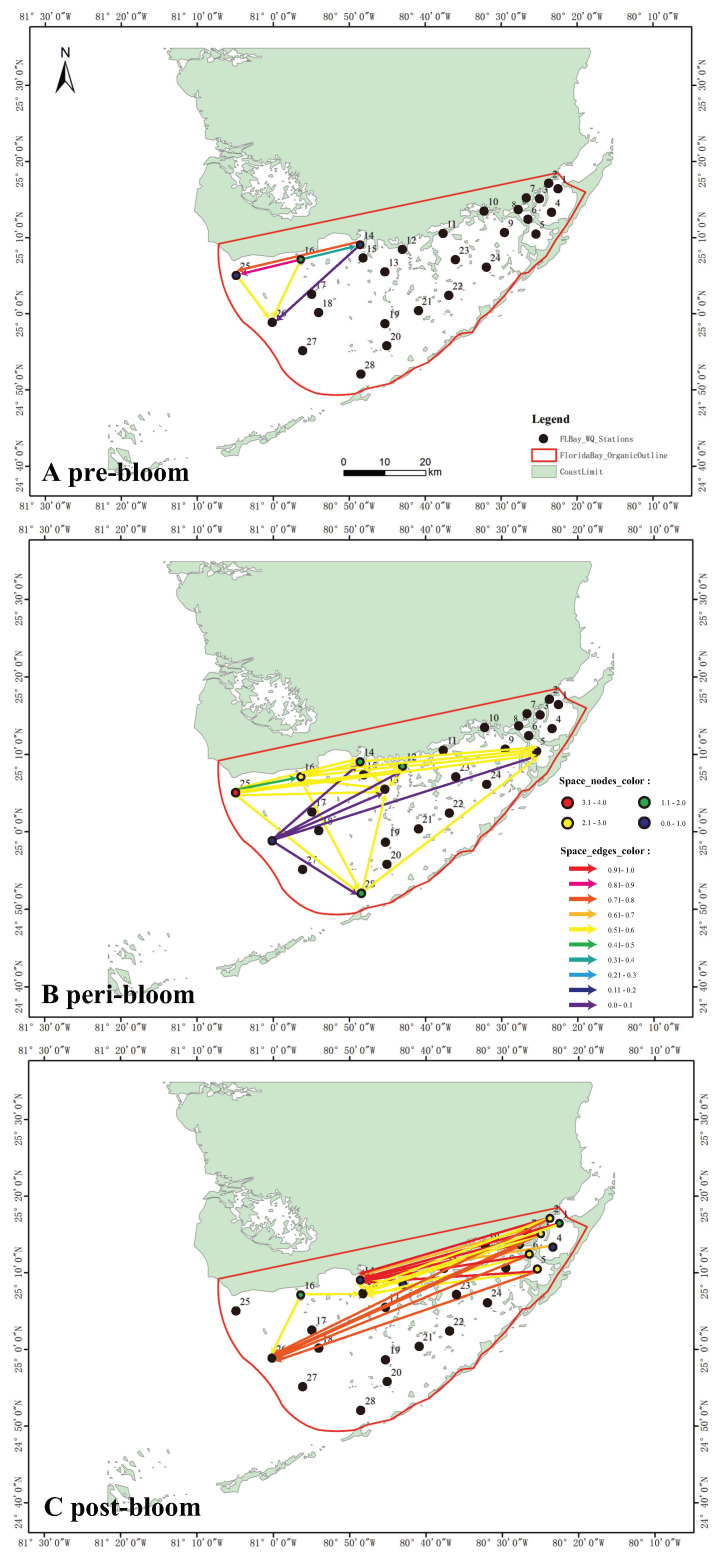
Inferred spatial CHLa for the 2004, 2005 and 2006 pre-, peri- and post-bloom periods in Florida Bay. Link and node color (from blue to red) is proportional to mTE based on CHLa interdependence between node pairs and *OTE* considering only $TE_{\mathrm{CHLa}->Env}$ where Env stands for all other environmental factors. East to west node and link dynamic increases are observed from 2004 to 2006 as well as a spreading network transition from regular/small-world to scale-free and regular (or uniform) with long-range connections for 2004, 2005 and 2006 (**A**–**C**). Each year corresponds to a different bloom precursor area and environmental factors (the central and northwest areas more affected by nutrients), widespread and extremely localized outbreak (the northeast more affected by temperature and turbidity and sequential effects of spreading).

**Figure 4 entropy-25-00636-f004:**
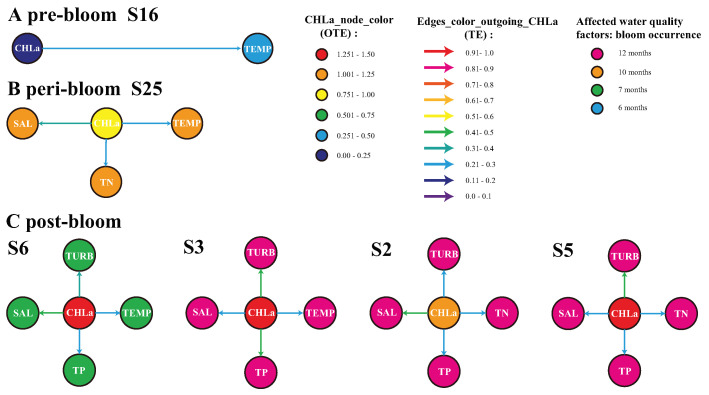
Inferred biogeochemical networks for the 2004, 2005 and 2006 pre-, peri- and post-bloom periods in Florida Bay. The purpose was to quantify local eco-environmental impacts for bloom sources. Only four nodes in 2006 and one node for 2004 and 2005 were considered because those are the most active in terms of the CHLa *OTE*. However, blooms are spreading phenomena, and other nodes are involved. Stations 16 and 25 are characterized by mangrove habitats in the west region, while stations 2, 3, 5 and 6 (displayed proportionally to a gradient of potential impact of CHLa on the environment) are characterized by coastal marshes and marine flat habitats in the east region of Florida Bay. The color of the directed edges is proportional to ranges of mTE for $TE_{\mathrm{CHLa}->Env}$ only. The node color for CHLa is proportional to *OTE* and, for other water quality factors, depends on the frequency of the local blooms during that year (manifesting the potential impact of CHLa on the environment): specifically, blue, green, orange and pink are for 6, 7, 10 and 12 months of bloom occurrence.

**Figure 5 entropy-25-00636-f005:**
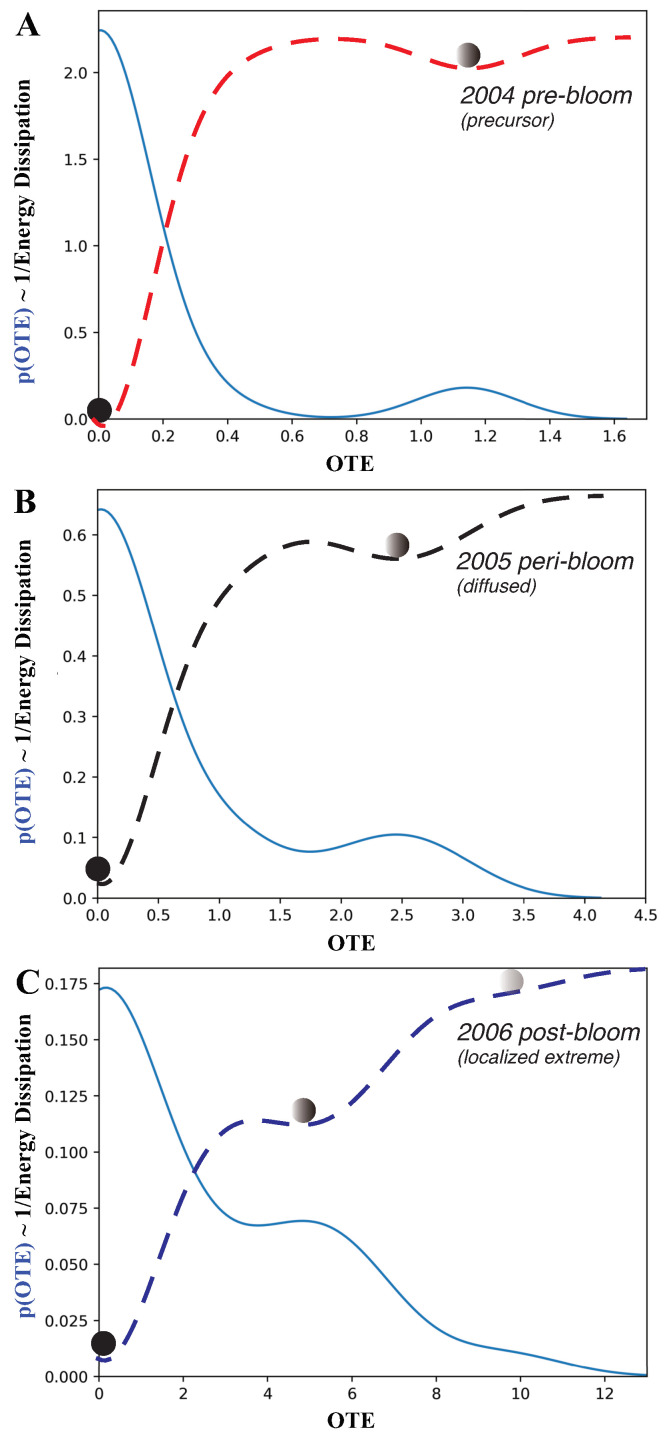
Probability distribution of the CHLa collective influence and ecosystem potential. (**A**–**C**) are for the 2004, 2005 and 2006 pre-, peri- and post-bloom periods in Florida Bay. CHLa’s collective influence was assessed based on *OTE* range and distribution, where the latter defines the energy potential (in dashed red, black and blue for 2004, 2005 and 2006 aligned with the distinct epidemic, transitory and endemic dynamics as in Figure 1B), stability of ecosystem states and transition probabilities from one to another.

## Data Availability

Data of Florida Bay blooms are available from Nelson et al. (2017), Mar. Ecol. Prog. Ser., that originally published the data.

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
