# Peer review of "Algal Bloom Ties: Spreading Network Inference and Extreme Eco-Environmental Feedback"

_entropy, 2023, doi:10.3390/e25040636_

Round 1

Reviewer 1 Report

The paper is interesting and well done and, apart from some format incorrectness, especially in the placement of citations, does not require major interventions. To make the reading more fluent, I would recommend reducing the use of acronyms, when the discourse allows, and I would recommend being more comprehensive in describing methods.

Line 80 – CHLa, for the first time it appears, change to: chlorophyll a (CHLa)

Line 87 – Change “FL”, in “Florida”

Line 94 – delete internal brackets

Lines 121-136 – the quotations Convertino et al. (2021), Li and Convertino (2021a), Galbraith and Convertino(2021), etc. are too many (as well as poorly placed) and should be focused

Line 142 – “.. egularly experiences algal blooms Phlips et al. (1999) as ..”, change in:  “.. egularly experiences algal blooms (Phlips et al., 1999) as ..”

line 175 – please specify pdfs

lines 233-234 – please specify acronyms TEGNN and OIF

Reviewer 2 Report

The authors presented an interesting application of mathematical methods for the assessment of blooms, but the article has serious shortcomings (e.g. sea currents are not included) and numerous errors and is not suitable for publication in its current form.

The article requires linguistic correction (numerous grammar and spelling errors).

The abstract should be thoroughly edited and shortened (e.g. remove unnecessary repetitions, there are no conclusions).

L 59 - Remove repeated "widespread"

L133, L 142 - Put the quoted authors in brackets

Section 1 is missing the purpose of the work

Fg. 2 - Improve readability

L153-156 - Provide the methodology for measuring these parameters?

L156-159: Move the sentence: CHLa has often been… to section 1.

L193: Move the sentence: In this paper we specifically…. to section 1.

Is there any information on sea currents in this body of water?

The following two issues reduce the scientific quality of this article (the authors misinterpret certain phenomena):

1 - Explain how CHLa bloom affects water temperature (it is surely the other way around).

2 - Similarly, the question of the effect of CHLa blooms on salinity - maybe the influx of salt water from the sea or fresh water from land influenced changes in CHLa concentration?

L394 - what is this sentence for?,

Section 4 - to be shortened and redrafted - as it stands, these are not conclusions.

Round 2

Reviewer 1 Report

Scientifically, I have no objection and find the work interesting.

For the format, I must point out the Journal requires that citations, in the text, follow an increasing numbering, while in the paper they are put with surnames and date. But, in References, the list shows the citations with the author's first name, with no numerical list (already missing in the text, as I said), which makes it difficult and laborious to compare between text and References.

Author Response

we revised the references according to MDPI style

Reviewer 2 Report

The manuscript has been improved as the reviewer's suggestion and it is acceptable now.

Author Response

we further revised some English and corrected some mistakes